# On-chip coherent conversion of photonic quantum entanglement between different degrees of freedom

Lan-Tian Feng[1,2,*], Ming Zhang[3,*], Zhi-Yuan Zhou[1,2,*], Ming Li[1,2], Xiao Xiong[1,2], Le Yu[1,2], Bao-Sen Shi[1,2], Guo-Ping Guo[1,2], Dao-Xin Dai[3], Xi-Feng Ren[1,2] & Guang-Can Guo[1,2]

In the quantum world, a single particle can have various degrees of freedom to encode quantum information. Controlling multiple degrees of freedom simultaneously is necessary to describe a particle fully and, therefore, to use it more efficiently. Here we introduce the transverse waveguide-mode degree of freedom to quantum photonic integrated circuits, and demonstrate the coherent conversion of a photonic quantum state between path, polarization and transverse waveguide-mode degrees of freedom on a single chip. The preservation of quantum coherence in these conversion processes is proven by single-photon and two-photon quantum interference using a fibre beam splitter or on-chip beam splitters. These results provide us with the ability to control and convert multiple degrees of freedom of photons for quantum photonic integrated circuit-based quantum information process.

[1] Key Laboratory of Quantum Information, University of Science and Technology of China, CAS, Hefei 230026, China. [2] Synergetic Innovation Center of Quantum Information and Quantum Physics, University of Science and Technology of China, Hefei 230026, China. [3] State Key Laboratory for Modern Optical Instrumentation, Centre for Optical and Electromagnetic Research, Zhejiang Provincial Key Laboratory for Sensing Technologies, Zhejiang University, Zijingang Campus, Hangzhou 310058, China. * These authors contributed equally to this work. Correspondence and requests for materials should be addressed to D.-X.D. (email: dxdai@zju.edu.cn) or to X.-F.R. (email: renxf@ustc.edu.cn).

Compared with free space and fibre optics, photonic integrated circuits (PICs) have attracted considerable attention owing to their small footprint, scalability, reduced power consumption and enhanced processing stability; thus, many quantum computation and quantum simulation protocols have been realized on quantum PICs[1–9]. Regarding recent investigations of quantum PICs, polarization entanglement and path entanglement are usually used. Polarization entanglement is quite important because the polarization of photons is easy to control in free space, and various proof-of-principle demonstration of quantum computation schemes were carried out based on the polarization degree of freedom[10–12]. For the path-encoding method, it is possible to establish a higher-dimensional Hilbert space[5], which means that we can encode more information per photon and increase the security of quantum systems[13–16]. However, the path-encoding method is not easy to scale up for practical implementations.

Recently, transverse waveguide modes were introduced as a new information encoder, and they were used in multi-core[17] and few-mode fibres[18], as well as in classical PICs[19], to satisfy the increasing demand for the capacity of optical interconnects. A multi-mode waveguide can support many transverse waveguide modes, which form a set of orthogonal basis for the transverse spatial distribution of energy; thus, it is appropriate for carrying more information using photons. For example, the eight-channel information-encoding process has been demonstrated in a 2.363-μm wide multi-mode waveguide[20]. This degree of freedom may have great potential in quantum optics, such as realizing high-dimensional quantum operation, maintaining the polarization entanglement resource in a high-birefringence integrated device.

We can use multiple degrees of freedom of a quantum particle simultaneously, which will certainly increase the information capacity of qubits[21–24]. As for photons, polarization, frequency, time, orbital angular momentum and even the transverse-mode entanglement have been used in free-space quantum systems, and some of them have been used in fibre quantum systems[25]. For example, quantum teleportation of the composite quantum states of a single photon encoded in both spin and orbital angular momentum was recently demonstrated in free-space[26]. Experimental realization of multi-degree-of-freedom entanglement poses significant challenges to the coherent control of multiple degrees of freedom simultaneously and to realizing quantum logic gates between independent qubits of different degrees of freedom.

In the following, to show the potential utility of transverse waveguide modes in the quantum information process, we demonstrate that quantum coherence is preserved when photons in different transverse waveguide modes propagate in a multi-mode waveguide. We will also show the coherent on-chip conversion of quantum states between different degrees of freedom, such as path, polarization and the transverse waveguide mode. Here coherent conversion refers to the preservation of coherence of quantum state, including the indistinguishability between the single photons, the stability of relative phase of superposition state and entangled state in the processes of photon transmission and conversion between different degrees of freedom.

## Results

**Experimental set-up**. The transverse waveguide modes discussed in this study are the three lowest-order modes, that is, $TE_0$, $TE_1$ and $TM_0$ in a multi-mode waveguide, as shown in upright inset of Fig. 1a. Here we use a silicon-on-insulator (SOI) strip waveguide with a cross-section of $\sim 750 \times 220$ nm. To effectively and accurately excite and manipulate these transverse waveguide modes in this multi-mode waveguide, the low-loss and low-

crosstalk mode (de)multiplexer is one of the most important devices. In this work, we choose two different structures for the conversion between different degrees of freedom. The first one is a special polarization-dependent mode converter, which can convert laser beam in TE and TM polarizations into the $TE_0$ and $TE_1$ modes[28,29], respectively, as shown in Supplementary Fig. 1a. The other one is a mode multiplexer, which can convert laser beam in path 1 and path 2 into the $TE_0$ and $TE_1$ waveguide modes[27], respectively, as shown in Supplementary Fig. 1b. Although these elements work well for laser beam, they have not yet been used for quantum signals. After conversion, the $TE_0$- and $TE_1$-mode photons propagate in the multi-mode waveguide for a certain distance and then convert back to photons with different polarizations or different optical paths, as shown in Fig. 1b.

To test the coherent property of the two photons after undergoing different conversion processes, a Hong–Ou–Mandel (HOM) interferometer is used. HOM interference is a basic type of quantum interference that reflects the bosonic properties of a single particle and is generally used to test the quantum properties of single qubits[30]. It can be described as follows: when two indistinguishable photons enter a 50/50 beam splitter (BS) from different sides at the same time, the two photons will come out together and never be in different output ports. Experiments typically control the arrival time of two photons by adjusting the path-length difference between them and measure the photon coincidence (the case in which two photons arrive at two detectors simultaneously) of the two output ports of the BS. When two indistinguishable photons completely overlap at the BS, they give rise to the maximum interference effect and no coincidence exists. Visibility is defined as $V_1 = (C_{max} - C_{min})/C_{max}$, where $C_{max}$ is the maximum coincidence and $C_{min}$ is the minimum coincidence. For perfect quantum interference, $C_{min} = 0$ and $V_1 = 1$. Or we can collect the photons from one output port of the BS, send them to the second 50/50 BS, and then measure the coincidence[31]. In this case, a peak will be observed and the visibility is modified as follows: $V_2 = (C_{max} - C_{min})/C_{min}$. For perfect quantum interference, $C_{max} = 2C_{min}$ and $V_2 = 1$.

The experimental set-up is shown in Fig. 2. The degenerate 1,558-nm photon pair source is generated using a type II phase-matched periodically poled potassium titanyl phosphate ($KTiOPO_4$) crystal in a Sagnac interferometer pumped by a continuous-wave 779-nm laser[32], which operates in a single circulation direction. Each photon is coupled into a single-mode fibre and then sent to a fibre array. In one arm, the fibre coupler is mounted on a one-dimensional translator with a step of 10 μm. By moving the translator, we can modify the arrival time of the single photons on BS and thus observe the HOM interference effect. The grating coupling method is used to couple the single photons into/out of the chip from/into the fibre arrays. The output photon pairs collected by the second fibre array are sent into a fibre BS to produce the HOM interference.

The indistinguishability of the photon pairs from the source is characterized using a standard HOM interferometer with a fibre BS. The dip represents the quantum interference of two photons, and the coherence length of the photons determines its width, as shown in Supplementary Fig. 2. Here we obtain a raw visibility of $96.3 \pm 2.8\%$ ($96.8 \pm 2.8\%$ with background subtraction) and an optical coherence length of $448.7 \pm 19.8$ μm. The deviation of the visibility from 100% is attributed to the polarization distortion of the photons during the propagation in the fibre, the photon source variability or both.

**Single-photon state conversion**. The coherent conversion of single-photon state between different degrees of freedom is tested at first. The sketch map and charge-coupled device picture of the

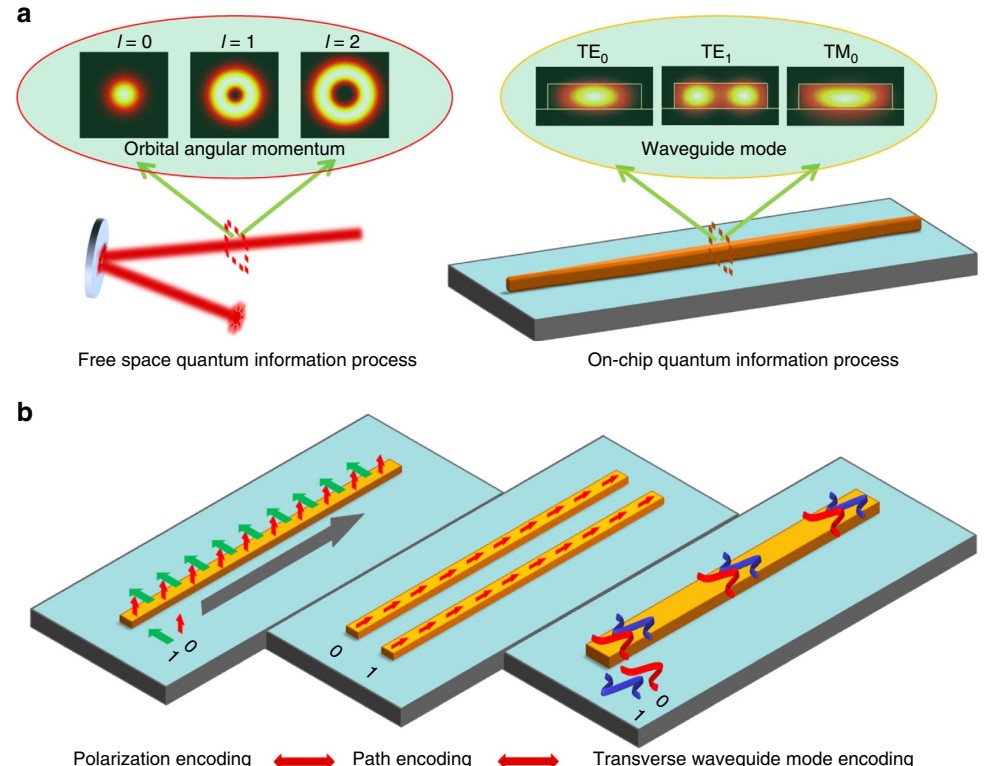

**Figure 1 | On-chip coherent conversion of photonic quantum entanglement between different degrees of freedom.** (**a**) Transverse waveguide mode as a new on-chip quantum information encoder. In free-space high-dimensional quantum information processes, orbital angular momentums of photons are usually used to encode information. Correspondingly, transverse waveguide mode can be used as a new degree of freedom for on-chip high-dimensional quantum information process. Inset on upright corner shows the energy distributions of the fundamental mode ($TE_0$ and $TM_0$) and the first higher-order mode ($TE_1$) in a multi-mode waveguide. The SOI strip waveguide has a cross-section of $\sim 750 \times 220$ nm. (**b**) On-chip coherent conversion of quantum states between different degrees of freedom, such as path, polarization and the transverse waveguide mode, is essential for using different degrees of freedom simultaneously.

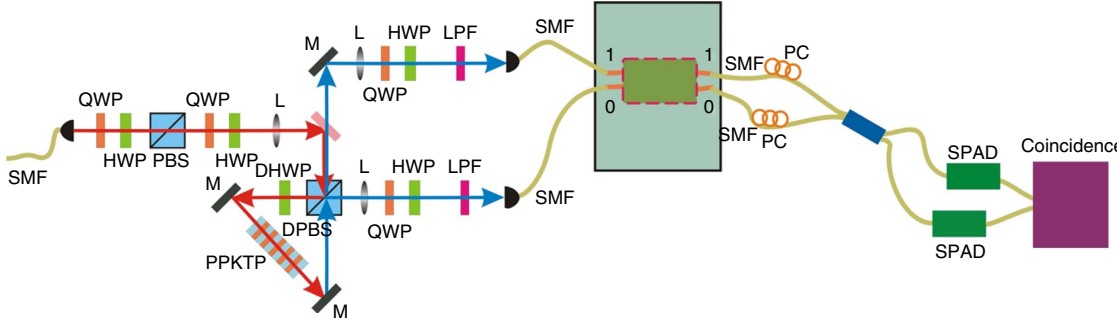

**Figure 2 | Experimental set-up for the two-photon source and sample measurement.** The continuous-wave pump laser at 779 nm is from a Ti:sapphire laser (Coherent MBR 110). It is collected into single-mode fibre (SMF) before entering the Sagnac-loop. A quarter wave plate (QWP) and a half wave plate (HWP) are used to control the phase and intensity of the pump beams in the Sagnac-loop. In the present experiment, the pump laser with vertical polarization is focused by a lens (L) with a focus length of 200 mm, whose beam waist is $\sim 40\,\mu m$ at the centre of the PPKTP crystal. The type II PPKTP (Raicol crystals) crystal has a size of $1 \times 2 \times 10$ mm, with a periodical poling period of $46.2\,\mu m$. The temperature of the PPKTP crystal is controlled by a home-made temperature controller with a stability of 2 mK. After a double PBS (DPBS), the polarization of the pump beam is changed to horizontal by a double HWP (DHWP) before the PPKTP crystal. The orthogonal polarized photon pairs generated in the counterclockwise direction are separated by the DPBS and collected into SMFs by using a lens set consisting of two lenses with different focus lengths of 100 and 50 mm at each output port of the DPBS, respectively. The pump beam is removed using a long pass filter (LPF, FELH1400). We use HWPs and QWPs to control the polarizations of the photon pair before injecting into the silicon chip. The output photons from the chip are detected by two InGaAs single-photon avalanche detectors (SPAD, D220, free running single-photon detector), with polarization controlled by fibre polarization controllers (PCs). The grating coupling method is used to couple the single photons into/out of the chip from/into the fibre arrays.

first sample are shown in Fig. 3a,b, respectively (see Supplementary Fig. 3 for detail). Two single photons with orthogonal polarizations from the fibre array are coupled into different single-mode waveguides, that is, the $TE_0$ and $TM_0$ mode,

by a TE-type grating and a TM-type grating, respectively. Then, the two single-mode waveguides combine together with a polarization BS (PBS) based on a bent directional coupler[28]. A special mode converter based on an adiabatic taper is cascaded

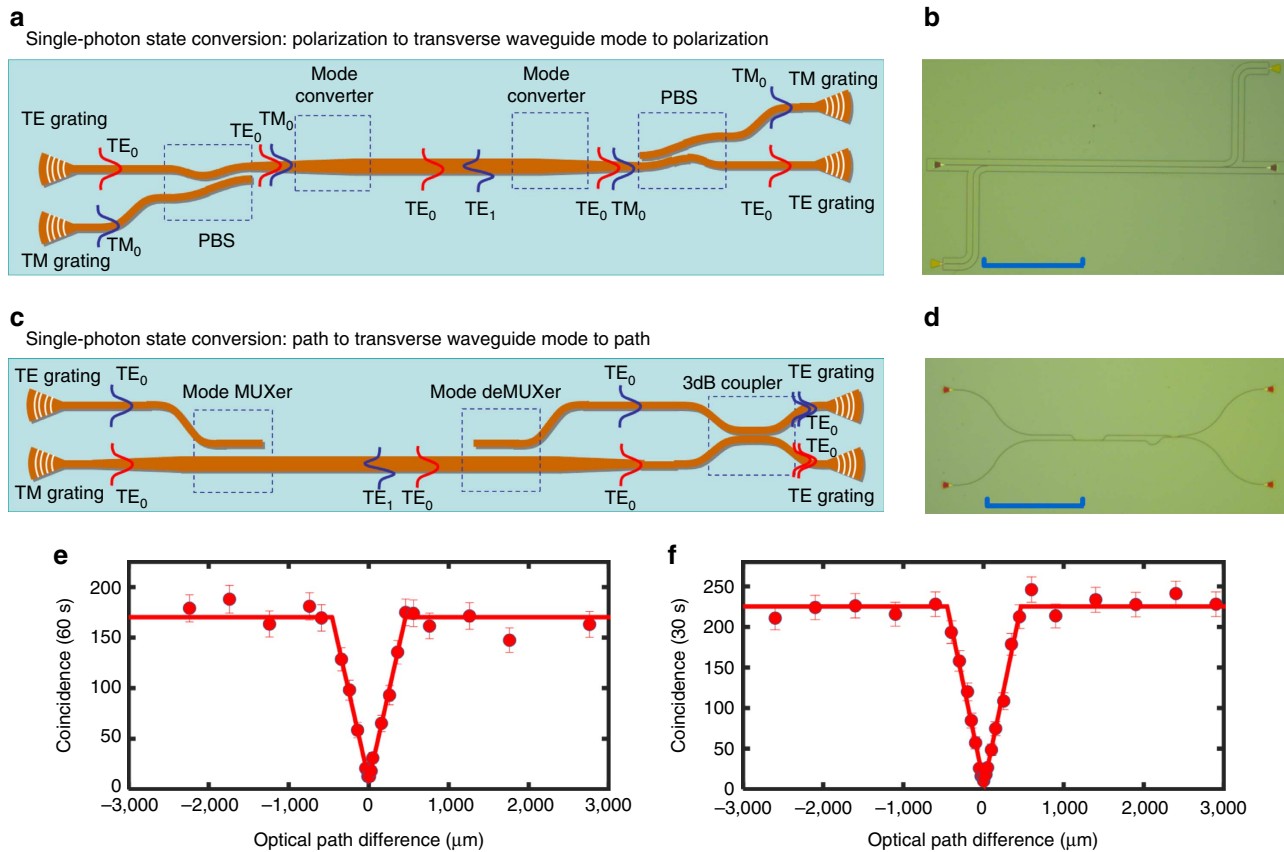

**Figure 3 | Single-photon state conversion.** (**a,b**) The sketch map and CCD (charge-coupled device) picture (scale bar, 250 μm) of the first sample, respectively. Two single photons with orthogonal polarizations from the fibre array are coupled into different single-mode waveguide as $TE_0$ and $TM_0$ modes, respectively, by a TE-type grating and a TM-type grating and then converted to different transverse waveguide modes, that, the $TE_0$ and $TE_1$ modes, by a mode converter after a PBS. After propagating along the multi-mode waveguide for 870 μm, the two photons are converted back with different polarizations and are coupled out to a fibre BS to produce a HOM interference. Coincidence measurement is performed when adjusting the position of the one-dimensional translator. (**c,d**) The sketch map and CCD picture (scale bar, 250 μm) of the second sample, respectively. Two single photons with the same polarization from the fibre array are coupled into different single-mode waveguides, both having the $TE_0$ mode, by two TE-type gratings, respectively. Then, with a mode multiplexer, photons in different optical paths are converted to different transverse waveguide modes. After propagating along the multi-mode waveguide for 30 μm, the two photons are divided into two distinct single-mode waveguides by a reversed process. HOM interference occurs on an on-chip BS (3-dB coupler), which was performed on the path degree of freedom. (**e**) HOM interference between the two photons, which undergo different conversion processes via a fibre BS for the first sample. The raw visibility is 92.3 ± 5.0% (94.8 ± 5.0% with background subtraction), which proves unambiguously the preservation of quantum coherence in the process of polarization to transverse waveguide mode and back to polarization. (**f**) HOM interference between two photons undergoing different conversion processes using an on-chip BS for the second sample. The visibility is 96.0 ± 3.3% (97.3 ± 3.3% with background subtraction), which proves unambiguously the preservation of quantum coherence in the process of path to transverse waveguide mode and back to path. Error bar comes from the Poisson statistical distribution. The experimental data are fitted with a triangle function.

and polarization-dependent mode conversion happens, as reported previously[29]. The $TM_0$ mode is then converted into the $TE_1$ mode after propagating along this adiabatic taper while there is no mode conversion for the $TE_0$ mode. As a result, photons with different polarizations are converted to different transverse waveguide modes (that is, the $TE_0$ and $TE_1$ modes). This is experimentally proven by a near-field scanning optical microscope, which can measure the evanescent field distribution of the guided mode in an optical waveguide. The results are shown in Supplementary Fig. 4, which clearly prove the function of the PBS and the mode converter. The two photons in the $TE_0$ mode and the $TE_1$ mode propagate along the multi-mode waveguide for 870 μm and are converted back to different polarizations by a similar mode converter structure. We collect the two output photons from the two different gratings and then send them to the HOM test system. By moving the one-dimensional translator, we can control the arrival time difference between the two photons and thus get the relation between the

coincidence and the path length difference. Figure 3e gives the measured HOM interference between the two photons undergoing different conversion processes. The raw visibility is 92.3 ± 5.0% (94.8 ± 5.0% with background subtraction) and the coherent length is 458.7 ± 37.8 μm, which proves unambiguously the preservation of quantum coherence during the conversion from the polarization to the transverse waveguide mode and back to polarization process.

Then, we tested the single-photon state conversion between the optical path and transverse waveguide-mode degrees of freedom with sample 2, as shown in Fig. 3c,d (see Supplementary Fig. 5 for detail). In this case, two photons with the same polarization from the fibre array are coupled into different single-mode waveguides, both having the $TE_0$ mode, by two TE-type gratings, respectively. Then, the two single-mode waveguides combine together with a mode multiplexer such that one $TE_0$ mode is converted to the $TE_1$ mode in the bus waveguide while the other $TE_0$ mode is kept unchanged. Thus, photons in different optical paths are converted

to different transverse waveguide modes. The two photons in the $TE_0$ mode and the $TE_1$ mode propagate along the multi-mode waveguide for 30 μm, and then are converted back to the $TE_0$ mode in two distinct single-mode waveguides with a mode demultiplexer. Unlike the first sample, HOM interference occurs on an on-chip BS (3-dB coupler). Figure 3f gives the result with a raw visibility of $96.0 \pm 3.3\%$ ($97.3 \pm 3.3\%$ with background subtraction), and the coherent length is $460.2 \pm 28.1$ μm, which illustrates the coherent conversion of quantum signals from path encoding to transverse waveguide-mode encoding.

**Quantum entanglement conversion.** Coherent conversion of the quantum-entangled state is also proven in our experiment using the third and fourth samples shown in Fig. 4a–d (see Supplementary Figs 6 and 7 for detail), respectively. For the third sample, two single photons with the same polarization are coupled to single-mode waveguides by two gratings, respectively. Then, they interfere at the BS and generate a two-photon quantum NOON state[33], encoded on the path as $(|2\rangle_0|0\rangle_1 - |0\rangle_0|2\rangle_1)/\sqrt{2}$, where $|n\rangle_i$ denotes $n$ photons in path $i$, for $n = 0, 1, 2$, and $i = 0, 1$. With a mode multiplexer, the state will be changed to $(|2\rangle_{TE_1}|0\rangle_{TE_0} - |0\rangle_{TE_1}|2\rangle_{TE_0})/\sqrt{2}$ because the

photons in path 0 will be in the $TE_1$ mode while photons in path 1 will be in the $TE_0$ mode. After propagation in the multi-mode waveguide for a distance of 30 μm, this transverse waveguide-mode two-photon NOON state is changed back to a path NOON state. Two-photon interference, or two-photon NOON state interference, is measured by using the second on-chip BS, as shown in Fig. 4e (red dots). The phase between the two arms is adjusted by using a thermal-tuning method. Classical interference is also measured for comparison with a visibility of $99.9 \pm 7.8\%$ (black dots). We observe a entangled state interference visibility of $90.3 \pm 7.8\%$ ($94.0 \pm 8.2\%$ with background subtraction) with a period (heater power $32.5 \pm 0.7$ mW) approximately half of the classical interference (heater power period is $66.8 \pm 2.8$ mW).

Finally, to show that these conversion processes can be cascaded, we combine several structures together on a single chip, and a two-photon quantum NOON state is converted between these three degrees of freedom. The sample is shown in Fig. 4c,d. Two single photons with the same polarization are coupled to single-mode waveguides and then interfere at the on-chip BS, generating a quantum-entangled state encoded on the path as $(|2\rangle_0|0\rangle_1 - |0\rangle_0|2\rangle_1)/\sqrt{2}$. It is first converted to a transverse

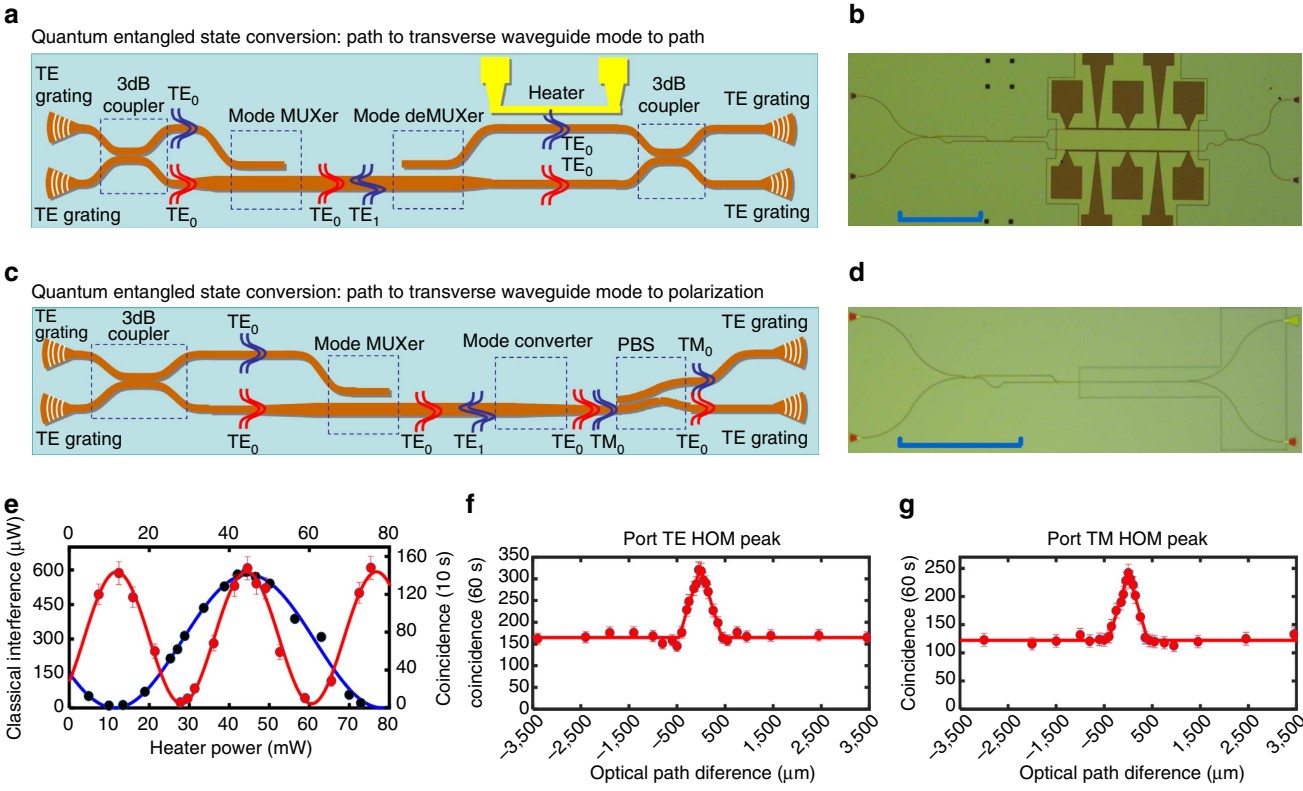

**Figure 4 | Quantum-entangled state conversion.** (**a,b**) The sketch map and CCD (charge-coupled device) picture (scale bar, 250 μm) of the third sample, respectively. Two single photons with the same polarization are coupled to single-mode waveguides by two gratings, respectively. Then, they interference at the first BS (3-dB coupler) and generate a two-photon quantum NOON state, encoded on the path as $(|2\rangle_0|0\rangle_1 - |0\rangle_0|2\rangle_1)/\sqrt{2}$. With a mode multiplexer, the state will be changed to $(|2\rangle_{TE_1}|0\rangle_{TE_0} - |0\rangle_{TE_1}|2\rangle_{TE_0})/\sqrt{2}$. After propagation in the multi-mode waveguide for a distance of 30 μm, this transverse waveguide-mode NOON state is changed back to a path NOON state. Two-photon interference, or two-photon NOON state interference, is measured by using the second on-chip BS, as shown in **e** (red dots). Classical interference is also measured for comparison (black dots). We observe a raw interference visibility of $90.3 \pm 7.8\%$ ($94.0 \pm 8.2\%$ with background subtraction) with a period approximately half of the classical interference. (**c,d**) are the sketch map and CCD picture (scale bar 250 μm) of the fourth sample, respectively. A quantum two-photon NOON state is generated by an on-chip BS and then converted between the three degrees of freedom (path, transverse waveguide mode and polarization). (**f,g**) HOM interference patterns between the two photons from TE and TM outputs of the sample, respectively. Error bar comes from the Poisson statistical distribution. The raw visibilities are $96.8 \pm 7.8\%$ ($98.2 \pm 7.9\%$ with background subtraction) and $96.7 \pm 8.3\%$ ($98.3 \pm 8.5\%$ with background subtraction), respectively. These results prove unambiguously the preservation of quantum coherence in the conversion of quantum-entangled state between different degrees of freedom. It should be noted that all the measurements of the HOM interference were performed on the path degree of freedom for the sake of simplification.

waveguide-mode NOON state $(|2\rangle_{TE_1}|0\rangle_{TE_0} - |0\rangle_{TE_1}|2\rangle_{TE_0})/\sqrt{2}$ and then changed to $(|2\rangle_{TM_0}|0\rangle_{TE_0} - |0\rangle_{TM_0}|2\rangle_{TE_0})/\sqrt{2}$, which is a polarization NOON state. To show that the whole process is actually the same as what we described above, it is important to show that the two photons at the output are either both in $H$ polarization or in $V$ polarization and never have different polarizations. We collect the output photons from each grating coupler and conduct the HOM interference test with a fibre BS. As shown in Fig. 4f,g, peaks are observed as the two photons arrive at the on-chip BS simultaneously. The raw visibilities are $96.8 \pm 7.8\%$ ($98.2 \pm 7.9\%$ with background subtraction) and $96.7 \pm 8.3\%$ ($98.3 \pm 8.5\%$ with background subtraction), and the coherent lengths are $446.1 \pm 37.0$ and $407.7 \pm 31.2\,\mu m$, respectively. This means that when the two-photon path NOON state is generated, there are two photons at output 1 or at output 2, as we predicted.

## Discussion

We conclude that our experiment demonstrates unambiguously the coherent propagation of quantum signals encoded on transverse waveguide modes and the on-chip coherent conversion of quantum entanglement between different degrees of freedom. Although only two lower transverse waveguide modes are discussed, this newly introduced degree of freedom shows us the possibility of encoding quantum information within a higher-dimensional Hilbert space, which is useful for the investigation of the on-chip high-dimensional quantum information process, such as teleportation using qudits[13,14], quantum dense coding[15] and quantum key distribution[16]. For example, a two-photon three-dimensional (3D) path-entangled state can be converted to a 3D transverse waveguide-mode-entangled state easily using our mode converters (similar demonstration between path and orbital angular momentum was recently realized in free space[34]).

The on-chip coherent conversion of quantum-entangled state that is encoded onto path, polarization and the transverse waveguide mode, shows us the ability to control these degrees of freedom, which has great potential in on-chip hyper-entangled quantum systems. Hyper-entanglement[21,35], where qubits are entangled in two or more degrees of freedom, has shown advantages in quantum information applications. Using hyper-entanglement will make it much easier to perform quantum logic gates[36], and will also enable new capabilities in quantum information process, such as remote preparation of entangled states, full Bell-state analysis and improved super-dense coding[22,23], as well as the possibility of quantum communication with larger alphabets[24]. Path-polarization hyper-entangled and cluster states of photons on a chip were recently realized[37].

The chips we used are based on SOI waveguides, which have been developed well and used widely because of the complementary metal oxide semiconductor (CMOS) compatibility and the ultra-small footprint. The operation wavelength of the silicon chip lies in the whole telecom band, which is compatible with present fibre communication networks. Quantum information processing based on silicon photonic chip will be a good candidate of quantum processor for quantum communication networks. Note that because different transverse waveguide modes have different effective refractive indices, relative phases will be generated when photons in different transverse waveguide modes propagate along the multi-mode waveguide. For the on-chip quantum information process, this phase can be adjusted by using a thermal-tuning method. In our experiment, all the measurements of the HOM interference were performed on the path degree of freedom for the sake of simplification.

While this report was being written, quantum interference between transverse waveguide modes was realized[38].

## Methods

**Sample design and fabrication.** The chip includes some key components, including the PBSs, the mode multiplexers and the mode converters. All the components are designed according to the optical waveguide theory and the coupled-mode theory. The simulation tools include the Fimmprop (PhotoDesign, Oxford, UK) using an eigenmode expansion and the matching method, and Lumerical software (Lumerical Solutions, Inc., London, UK) with the 3D time-domain finite-difference method. The PBS is designed with a bent directional coupler consisting of two parallel bent waveguides[28]. These two bent waveguides have different core widths and could be designed to satisfy the phase-matching condition for the coupling of TM polarization; consequently, TM-polarized light could be coupled to the cross port completely when choosing the length of the coupling region appropriately. On the other hand, for TE polarization, the phase-matching condition is not satisfied because of the birefringence of the waveguides. Thus, TE-polarized light goes through without any significant coupling. In this way, TE- and TM-polarized lights are separated within a very short length that is close to the coupling length of TM polarization. The mode multiplexer is designed with an asymmetric directional coupler, which consists of a narrow access waveguide close to the wide bus waveguide[27]. The widths of the narrow access waveguide and the wide bus waveguide are chosen optimally to satisfy the phase-matching condition so that the fundamental mode of the narrow access waveguide can be coupled to the first higher-order mode in the bus waveguide completely. On the other hand, there is almost no coupling from the fundamental mode of the wide bus waveguide to any modes in the narrow access waveguide. The mode converter is designed with an adiabatic taper based on SOI strip waveguides with an air upper-cladding[29]. For an SOI strip waveguide whose cross-section has vertical asymmetry, mode hybridization happens when choosing some special core width $w_{co0}$. The hybridized modes have comparable $x$ and $y$ components for the electrical fields. Such mode hybridization will introduce a mode conversion when light propagates along an adiabatic taper structure whose end widths $w_1$ and $w_2$ are chosen such that $w_1 < w_{co0} < w_2$.

For the fabrication of the present sample, the process started from an SOI wafer with a 220-nm-thick top silicon layer. An E-beam lithography process with the MA-N2403 photoresist was carried out to make the pattern of waveguides, which was then transferred to the top silicon layer via an inductively coupled-plasma-etching process. Grating couplers were made using a second etching process to achieve an efficient fibre-chip coupling.

**Grating coupling method.** Grating couplers are very popular for realizing efficient coupling between the chip and fibres at the input/output ends[39]. Here we use two types of grating couplers, that is, TE type and TM type, which are designed for TE- and TM-polarized lights, respectively. In our design, the grating periods are chosen as 640 and 1040 nm for the TE- and TM-type grating couplers, respectively. To avoid reflection at the waveguide-grating interface, light is coupled in (out) at a small angle (15° in our experiment) with respect to the vertical direction. The peak coupling efficiencies are about 30%.

**Photon source.** The continuous-wave pump laser at 779 nm is from a Ti: sapphire laser (Coherent MBR 110). It is collected into single-mode fibre before entering the Sagnac-loop. A quarter-wave plate and a half-wave plate (HWP) are used to control the phase and intensity of the pump beams in the Sagnac-loop. In the present experiment, the pump laser with vertical polarization is focused by a lens with a focus length of 200 mm, whose beam waist is $\sim 40\,\mu m$ at the centre of the periodically poled potassium titanyl phosphate (PPKTP) crystal. The type II PPKTP (Raicol crystals) crystal has a size of $1 \times 2 \times 10$ mm, with a periodical poling period of 46.2 μm. The temperature of the PPKTP crystal is controlled by a home-made temperature controller with a stability of 2 mK. After a double PBS (DPBS), the polarization of the pump beam is changed to horizontal by a double HWP before the PPKTP crystal. The orthogonal polarized photon pairs generated in the counterclockwise direction are separated by the DPBS and collected into single-mode fibres by using a lens set consisting of two lenses with different focus lengths of 100 and 50 mm at each output port of the DPBS, respectively. The pump beam is removed using a long pass filter (FELH1400). We use HWPs and quarter-wave plates to control the polarizations of the photon pair before injecting into the silicon chip. The output photons from the chip are detected by two InGaAs single-photon avalanche detectors (D220, free running single-photon detector).

**Data availability.** The authors declare that the data supporting the findings of this study are available within the article and its Supplementary Information files.

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

## Acknowledgements

This work was funded by NBRP (grant nos 2011CBA00200 and 2011CB921200), the Innovation Funds from the Chinese Academy of Sciences (grant no. 60921091), NNSFC (grant nos11374289, 61590932, 61422510, 11374263, 61431166001 and 61525504), the Fundamental Research Funds for the Central Universities, the Open Fund of the State Key Laboratory on Integrated Optoelectronics and the Doctoral Fund of the Ministry of Education of China (no. 20120101110094).

## Author contributions

All authors contributed extensively to the work presented in this paper; M.Z. and D.-X.D; prepared the samples; L.-T.F., Z.-Y.Z., B.-S.S. and X.-F.R. performed the measurements and data analysis; M.L., X.X., L.Y., G.-P.G. and G.-C.G. conducted theoretical analysis; X.-F.R. and D.-X.D. wrote the manuscript and supervised the project.

## Additional information

**Competing financial interests:** The authors declare no competing financial interests.

