## [Peer review file · Nature Communications]

Reviewers' Comments:

Reviewer #1 (Remarks to the Author)

The manuscript reports on experimental demonstration of integrated linear-optical waveguided device for manipulation of quantum states as an alternative to free-space configuration. The result is interesting for the general audience. The experimental verification has been performed with great degree of accuracy. This is clearly visible from the triangular shape of constructive and destructive interference features. However, several modifications must be considered before this paper could be published.

1) The issue of "coherent conversion" has been at the center of attention. However, the linear optical transformation described in this manuscript is supposed to be a unitary device and not supposed to affect the degree of spectral coherence. It is unclear what type of coherence the authors have in mind when claiming the "coherent conversion". Is this a spatial coherence of a single spatial mode versus the mixture of spatial mode while using the same frequency mode? This issue definitely requires clarification.

2) The set-up used for the quantum state verification is not really a Hong-Ou-Mandel interferometer because it is based on the use polarization and frequency entanglement of participating photons. This allows one to switch from dip to peak and to observe a triangular shape of the interference pattern [see Journal of the Optical Society of America, v. B12, p.859 (1995)] that is not possible with the original HOM configuration. This configuration is so different from the original HOM interferometer that it also allows to observe full quantum interference even when the photons come to the beamsplitter at very distinct times [see Physical Review Letters, v. 77, p.1917 (1996).]

3) The issue of subtracting the background is very important in all quantum optical applications. It is also important in evaluating the usefulness of the present device. The raw data must be reported throughout the manuscript with the original visibility before the background subtraction.

4) The use of the term "NOON state" is not justified in this presentation. The demonstrated design is only capable of operating with $|2\rangle|0\rangle - |0\rangle|2\rangle$ states. This state is a very special in quantum optics and is linked with the Mach-Zehnder interferometer dimensionality and topology. The presented system is not capable of working with N higher than 2. The generalization to the NOON state should be removed.

5) The language needs to be cleaned up. There are too many problematic expressions. For example: "Therefore, efficiently and fully controlling of those degrees of freedom simultaneously is on demanding."; "Interestingly, there is no limitation of using only one freedom." etc..

In conclusion, the manuscript reports on a very useful integrated device for linear-optical manipulation of entangled degrees of freedom on a chip. This paper could be suitable for publication after all abovementioned concerns have been addressed in revision.

Reviewer #2 (Remarks to the Author)

In the work by Feng and colleagues "On chip coherent conversion of photonic quantum entanglement between different degrees of freedom" the authors present a novel method to effectively convert and switch between path, polarization and transverse waveguide mode degrees of freedom of a two-photon quantum system by exploiting an integrated photonic device.

In their setup, the authors use a periodically poled crystal in a Sagnac-type source to generate photons entangled in polarization and over two different modes. The photons are then injected into a chip which, depending on its configuration, permits to convert path and polarization degree of freedom into transverse waveguide mode (TWM), identified by TE₀, TE₁, TM₀ and TM₁ and

viceversa. In order to achieve this results they use integrated TE and TM gratings which converts path to proper TWM, mode MUXers which allow to change selectively between mode TE0 and TE1, mode converters which allows to transform TM0 in TE1 and viceversa. The authors perform single photon characterization of the behaviour of their device and then test it in a two photon regime in which they prove entanglement conversion.

In the following I will present my main concerns about this work.

1. Main scope of the experiment: the authors should clarify why using TWM represents a useful resource for quantum information processes. For example, it would allow to maintain the polarization entanglement resource in a high-birifringence integrated device that would otherwise cancel it.
2. The authors state that "the results can be extended to other higher-order waveguide modes intuitively". While I believe that there are no conceptual limits for this, the part of the paper about swapping entanglement between different degrees of freedom depends strongly on the fact that it's very easy to create a $|20\rangle + |02\rangle$ state by using a simple Beam Splitter. (HOM). Increasing complexity by adding additional TWMs could be useful only if the generation of of general N-qubits N00N states is feasible, which it isn't at the current state of the art. The authors should elaborate on this regard by addressing scalability of their system.
3. The authors should better clarify with some detail the grating coupling method in order to make the manuscript more readable.
4. Page 8, when the authors define the N00N state, I believe the label of the kets should be 0,1 instead of 1,2 to match notation for the path modes.
5. In the discussion the authors discuss the possible connection of their work with hyperentanglement. Besides the paper of ref. [...] (Kwiat et al.), the author should make a reference to an other example of hyperentangled states realized with the path and the polarization of photons (PRL, 95, 240405 (2005)). Furthermore, a recent realization of hyperentangled states exploiting the same degrees of freedom within an integrated device should be also cited (Light: Science & Applications (2016) 5, e16064; doi: 10.1038/lsa.2016.64.)
6. Label of Figure 3, third row: TE0 and TM0 instead of TE0 and TE0
7. Figure 2 need labels for the experimental devices. I would add a label "0" and "1" over the two outputs of the DPBS which identify the path degree of freedom
8. The authors always demonstrate the presence of entanglement by measuring HOM like peak/dip. This means that ultimately they are performing a measurement on the path, as they are testing coherence between two photon entering in two ports of a BS. As a consequence, in the last figure, a) they are performing path->TWM->path (and they are detecting path coherence) and in figure b) they are performing path->TWM->polarization->path and measuring again path coherence by converting polarization in path with the integrated PBS. I think this should be specified both in the figure and in the main text.

In conclusion, the manuscript is technically and scientifically sound and the experimental results are unambiguous . This work is interesting and novel enough to warrant publication in Nature Communication, as it represents a significant technical tools that allowing to , 'convert back and forth between different degrees of freedom, once the previous concerns are addressed.

Responses to the comments from the first reviewer

General comment: The result is interesting for the general audience. The experimental verification has been performed with great degree of accuracy. In conclusion, the manuscript reports on a very useful integrated device for linear-optical manipulation of entangled degrees of freedom on a chip. This paper could be suitable for publication after all abovementioned concerns have been addressed in revision.

Our reply: We are grateful for the reviewer's conclusion that our result is interesting for the general audience and the manuscript is suitable for publication after revision. The responses to the reviewer's concerns are given below in detail.

Comment 1: The issue of "coherent conversion" has been at the center of attention. However, the linear optical transformation described in this manuscript is supposed to be a unitary device and not supposed to affect the degree of spectral coherence. It is unclear what type of coherence the authors have in mind when claiming the "coherent conversion". Is this a spatial coherence of a single spatial mode versus the mixture of spatial mode while using the same frequency mode? This issue definitely requires clarification.

Our reply: Thanks for the reviewer's advice. The "coherent conversion" mentioned in our manuscript referred to the preservation of coherence of quantum state (including the indistinguishability between the single photons, the stability of relative phase of superposition state and entangled state) in the process of photon transmission and conversion between different degrees of freedom.

To testify whether any de-coherence process occurred to the polarization, spectral and spatial mode degree of freedom in our mode converters and mode (de)multiplexers, the HOM interferences between two single photons were performed. The measured high visibility has verified the high indistinguishability of single photons, and proved that the temporal, polarization and spatial mode coherence of photons are preserved very well. In addition, the first order coherence has also been tested with a Mach-Zehnder interferometer, in which the stability of the phase in the conversion process was demonstrated.

In order to avoid any confusion, we have given some explanation about "coherent conversion" as suggested.

(Paragraph 2, Page 3):"... Here, coherent conversion referred to the preservation of coherence of quantum state, including the indistinguishability between the single photons, the stability of relative phase of superposition state and entangled state in the process of photon transmission and conversion between different degrees of freedom...".

Comment 2: The set-up used for the quantum state verification is not really a Hong-Ou-Mandel interferometer because it is based on the use polarization and frequency entanglement of participating photons. This allows one to switch from dip to peak and to observe a triangular shape of the interference pattern [see Journal of the Optical Society of America, v. B12, p.859 (1995)] that is not possible with the

original HOM configuration. This configuration is so different from the original HOM interferometer that it also allows to observe full quantum interference even when the photons come to the beamsplitter at very distinct times [see Physical Review Letters, v. 77, p.1917 (1996).]

Our reply: We agree with the referee that for the experimental setup used in (J. Opt. Soc. Am. B, 12, 859 (1995)), it is possible to switch from dip to peak and to observe a triangular shape of the interference pattern (J. Opt. Soc. Am. B, 12, 859 (1995)). We also agree that it is possible to observe the full quantum interference even when the photons come to the beamsplitter (BS) at very distinct times (Phys. Rev. Lett. 77, 1917 (1996)). In those works, polarization or frequency entanglement of participating photons were quite important. The photon pairs with different polarization generated from spontaneous parametric down conversion (SPDC) process arrive at the BS, and the t-t (two photons both transmitted the BS) and r-r (two photons both being reflected by the BS) cases correspond to the two terms of the Einstein-Podolsky-Rosen (EPR)-Bohm like states: $|HV\rangle \pm |VH\rangle$, where $|H\rangle$ and $|V\rangle$ represent horizontal and vertical polarization of photons, respectively. Therefore, rotating polarization analyzers before the detectors by different angles result in the polarization interference of this entangled state.

Our experimental setup is different from those mentioned above. In our experiment, though a Sagnac interferometer was used to produce photon pairs, we just used a single circulation direction, and the generated two photons were changed to the same polarization before they were coupled into the silicon chip. There is no polarization entanglement between the two photons and they were indistinguishable. Each photon was coupled into a single mode fibre and then the interference between the two photons was realized by a fiber BS or an on-chip BS. This experimental setup is similar with the original HOM interferometer (Hong, C. K., Ou, Z. Y., and Mandel, L. Measurement of subpicosecond time intervals between two photons by interference. Phys. Rev. Lett. 59, 2044 (1987)).

For the triangular shape of the interference pattern, it came from the spectral shape of the down-converted photons (see reference PRL, 94, 083601 (2005)). The peak interference pattern came from the modified HOM interferometer (see reference [31], J. Opt. Soc. Am. B, 6, 1221-1226(1989)), where we collected the two photons from one output port of the on-chip BS and measured the relation between the two-photon cases and arrival time difference using a fiber BS, as described in paragraph 1 on page 5.

We are sorry for the ambiguous description about our experimental setup, which may mislead the referee. In order to avoid any misleading, we have made some modifications about this in the revised version.

Comment 3: The issue of subtracting the background is very important in all quantum optical applications. It is also important in evaluating the usefulness of the present device. The raw data must be reported throughout the manuscript with the original visibility before the background subtraction.

Our reply: We agree with the referee that subtracting the background is very important in all quantum optical applications. In the revised version, we have provided the original data without the background subtraction for all the interference results as suggested.

Comment 4: The use of the term "NOON state" is not justified in this presentation. The demonstrated design is only capable of operating with $|2\rangle|0\rangle - |0\rangle|2\rangle$ states. This state is a very special in quantum optics and is linked with the Mach-Zehnder interferometer dimensionality and topology. The presented system is not capable of working with N higher than 2. The generalization to the NOON state should be removed.

Our reply: We agree with the referee and have removed the statement about the generalization to the NOON state in the revised manuscript.

Comment 5: The language needs to be cleaned up. There are too many problematic expressions. For example: "Therefore, efficiently and fully controlling of those degrees of freedom simultaneously is on demanding."; "Interestingly, there is no limitation of using only one freedom." etc..

Our reply: Thanks for the suggestion, we have re-examined the writing word by word, and cleaned up the language as best as we can.

Responses to the comments from reviewers 2

General comment: In conclusion, the manuscript is technically and scientifically sound and the experimental results are unambiguous. This work is interesting and novel enough to warrant publication in Nature Communication, as it represents a significant technical tools that allowing to convert back and forth between different degrees of freedom, once the previous concerns are addressed.

Our reply: We thank the referee's conclusion that our manuscript is technically and scientifically sound and the experimental results are unambiguous, and is interesting and novel enough to warrant publication in Nature Communication after reversion. The detailed responses to the mentioned concerns are given below.

Comment 1: Main scope of the experiment: the authors should clarify why using TWM represents a useful resource for quantum information processes. For example, it would allow to maintain the polarization entanglement resource in a high-birefringence integrated device that would otherwise cancel it.

Our reply: Thanks for this nice comment/suggestion. We have explained the advantage of using TWM in the Introduction section.

(Paragraph 1, Page 3): "This degree of freedom may have great potential in quantum optics, such as realizing high-dimensional quantum operation, maintaining the polarization entanglement resource in a high-birefringence integrated device."

Comment 2: The authors state that "the results can be extended to other higher-order waveguide modes intuitively". While I believe that there are no conceptual limits for this, the part of the paper about swapping entanglement between different degrees of freedom depends strongly on the fact that it's very easy to create a $|20\rangle+|02\rangle$ state by using a simple Beam Splitter. (HOM). Increasing complexity by adding additional TWMs could be useful only if the generation of general N-qubits $N00N$ states is feasible, which it isn't at the current state of the art. The authors should elaborate on this regard by addressing scalability of their system.

Our reply: We agree with the referee that our work of swapping entanglement between different degrees of freedom can hardly be used for the general N-qubits $N00N$ states. In our manuscript, the statement "the results can be extended to other higher-order waveguide modes" means is that two-photon higher-dimensional entanglement can be realized by using three or more waveguide modes. For example, a two-photon three dimensional path entangled state can be converted to a three-dimensional TWM entangled state easily using our mode converters (a similar demonstration on the high dimensional entanglement between path and orbital angular momentum in free space was recently reported in Nature Communications, 5 5502 (2014)). In order to clarify this, we have some comments as suggested.

(Paragraph 1, Page 10):"... For example, a two-photon three dimensional path entangled state can be converted to a three dimensional transverse waveguide mode entangled state easily using our mode

converters (a similar demonstration on the high dimensional entanglement between path and orbital angular momentum in free space was recently reported [34])...".

Comment 3: The authors should better clarify with some detail the grating coupling method in order to make the manuscript more readable.

Our reply: Thanks for the referee's advice. We have added a detailed description of the grating coupling method in the "Method" section of the revised manuscript.

(Paragraph 1, Page 13): "Grating couplers are very popular for realizing efficient coupling between the chip and fibers at the input/output ends [39]. Here we used two types of grating couplers, i.e., TE-type and TM-type, which are designed for TE- and TM-polarized lights, respectively. In our design, the grating periods are chosen as 640 nm and 1040 nm for the TE- and TM-type grating couplers, respectively. To avoid reflection at the waveguide-grating interface, the grating is detuned and light is coupled in (out) at a small angle (15° in our experiment) with respect to the vertical direction. The peak coupling efficiencies are about 30%."

[39]. F. Van Laere, T. Claes, J. Schrauwen, S. Scheerlinck, W. Bogaerts, D. Taillaert, L. O'Faolain, D. Van Thourhout, R. Baets. Compact focusing grating couplers for silicon-on-insulator integrated circuits. IEEE Photonics Technology Letters, 19(23), 1919-1921, 2007.

Comment 4: Page 8, when the authors define the N00N state, I believe the label of the kets should be 0,1 instead of 1,2 to match notation for the path modes.

Our reply: We are sorry for this mistake. It has been corrected in the revised version.

Comment 5: In the discussion the authors discuss the possible connection of their work with hyperentanglement. Besides the paper of ref. [...] (Kwiat et al.), the author should make a reference to another example of hyperentangled states realized with the path and the polarization of photons (PRL, 95, 240405 (2005)). Furthermore, a recent realization of hyperentangled states exploiting the same degrees of freedom within an integrated device should be also cited (Light: Science & Applications (2016) 5, e16064; doi: 10.1038/lisa.2016.64.)

Our reply: We appreciate the referee for introducing us these relevant references. In the revised manuscript, these papers have been included in the reference list as [35] and [37], respectively.

Comment 6: Label of Figure 3, third row: TE0 and TM0 instead of TE0 and TE0

Our reply: Thank the referee for carefully reading our manuscript and pointing out our mistake. We have corrected them in the revised version.

Comment 7: Figure 2 need labels for the experimental devices. I would add a label "0" and "1" over the two outputs of the DPBS which identify the path degree of freedom.

Our reply: As suggested, we have added labels "0" and "1" in Figure 2 to identify them.

Comment 8: The authors always demonstrate the presence of entanglement by measuring HOM like peak/dip. This means that ultimately they are performing a measurement on the path, as they are testing coherence between two photon entering in two ports of a BS. As a consequence, in the last figure, a) they are performing path->TWM->path (and they are detecting path coherence) and in figure b) they are performing path->TWM->polarization->path and measuring again path coherence by converting polarization in path with the integrated PBS. I think this should be specified both in the figure and in the main text.

Our reply: We agree with the referee that in our experiment, all the measurements of the HOM interference were performed on the path for the sake of simplification. According to this suggestion, this was clarified in the figure captions (Figures 3 and 4) and in the main text (paragraph 1, page 11) in the revised manuscript.

Fig. 3. "...HOM interference occurs on an on-chip BS, which was performed on the path degree of freedom."

Fig. 4. "... It should be noted that, all the measurements of the HOM interference were performed on the path degree of freedom for the sake of simplification."

(Paragraph 1, Page 11): "...In our experiment, all the measurements of the HOM interference were performed on the path degree of freedom for the sake of simplification."

Reviewers' Comments:

Reviewer #1 (Remarks to the Author)

The authors have addressed all and every of my original concerns in a very constructive way making the manuscript suitable for publication in Nature Communications in its current form.